# Surgical Treatment of Medication-Related Osteonecrosis of the Jaw: A Retrospective Study

**DOI:** 10.3390/ijerph17238801

**Published:** 2020-11-26

**Authors:** Na Rae Choi, Jung Han Lee, Jin Young Park, Dae Seok Hwang

**Affiliations:** Department of Oral Maxillofacial Surgery, School of Dentistry, Pusan National University, Yangsan 50612, Korea; nalaecism@naver.com (N.R.C.); omsljh@naver.com (J.H.L.); forfind@pusan.ac.kr (J.Y.P.)

**Keywords:** MRONJ, bisphosphonate, osteoporosis, surgical treatment

## Abstract

The purpose of this study was to confirm the success rate of surgical treatment of medication-related osteonecrosis of the jaw (MRONJ) in patients at a single institution (Association of Oral and Maxillofacial Surgery (AAOMS) stages 1, 2, or 3), and to identify the factors that influence treatment outcomes. As a result of analyzing the outcomes of treatment, surgical “success” was achieved in 93.97% (109) of cases, and “failure” was observed at 6.03% (7) cases. Analysis of patient factors that potentially affect treatment outcomes showed that zoledronate dose (*p* = 0.005) and the IV (intravenous) injection of drugs (*p* = 0.044) had significant negative impacts.

## 1. Introduction

Medication-related osteonecrosis of the jaw (MRONJ) was defined by the American Association of Oral and Maxillofacial Surgery (AAOMS) in 2014 as ischemic necrosis of the jaw caused by drugs that inhibit angiogenesis (antiangiogenic agents) or bone metabolism (antiresorptive agents), the latter is used to treat osteoporosis. The prevalence of MRONJ in the Korean population is around 0.04%, and its prevalence among tumor patients has been reported to be 1.86% [1].

Treatment strategies for MRONJ are controversial, especially for early-stage disease. AAOMS presented criteria for disease staging and treatment methods in a Position Paper released in 2014 [2]. When treating patients in stage 1 and 2, they recommended conservative treatment—taking antibiotics and trying gargle with 0.12% chlorhexidine first. When there is a concern about colonization of bacteria due to the formation of sequestra, a surgical procedure to selectively remove them was recommended. Many who agree with the treatment goals of AAOMS claim the effectiveness of conservative treatment [2,3,4,5,6,7]. However, many authors have recently supported the positive effect of resection of the affected area, which is expected to relieve symptoms quickly [4,8,9,10]. Hayashida et al. argued that sequestrum rarely returns to normal bone, and if non-surgical treatment is performed, continuous visits are required, and quality of life (QOL) rapidly declines [8]. Klingelhoffer et al. also argued that early surgical treatment can reduce lesion recurrence [10].

Although many studies have been performed on MRONJ treatment methods, no definite conclusion has been reached [1,3,4,5,11]. Therefore, this study was designed to provide information on the effectiveness of surgical treatment. More specifically, we investigated the success rates of surgical treatment in all stages of MRONJ and evaluated treatment outcomes using: (1) assessments of lesion exacerbation after treatment (defined as an AAOMS stage change), and (2) times to treatment termination.

## 2. Material and Methods

### 2.1. Patients

This retrospective study was approved by the Institutional Review Board of Pusan National University Dental Hospital (PNUDH-2019-034). It was performed for patients who received surgical treatment from January 2014 to December 2018, and it was continued after treatment at Pusan National University Dental Hospital Oral and Maxillofacial Surgery. They enrolled continuously during the period, but there was no regular periodicity. All patients had a history of medications (orally or intravenously) for osteoporosis. When collecting the medical history of these patients, the history of taking bisphosphonate (BPs) was focused. However, it was difficult to collect information on their BPs dosage. Information such as smoking status, oral hygiene, socioeconomic status, health status, and presence of co-morbidities was not collected. Table 1 shows the types of BPs taken by patients. The source of this table is J. K. Lee et al. [12]. Injectables were indicated as inj., and other drugs were PO.

The diagnostic criteria used and the staging of MRONJ were as recommended by AAOMS [2]. AAOMS said that MRONJ can be diagnosed in the following cases; (1) Current or previous treatment with antiresorptive or antiangiogenic agents; (2) Exposed bone or bone that can be probed through an intraoral or extraoral fistula(e) in the maxillofacial region that has persisted for more than eight weeks; and (3) No history of radiation therapy to the jaws or obvious metastatic disease to the jaws.

Exclusion criteria are as follows: (1) follow-up of less than 3 months and (2) history of steroid use. The total number of patients who satisfied the conditions was 116. Patients were prescribed antibiotics, that is, 500 mg of amoxicillin and chlorhexidine gargle (chlorhexidine gluconate solution 0.005 mL/g) 3 times a day routinely, and when pus was observed, 250 mg of metronidazole was added two weeks prior to surgical treatment. Antiresorptive and antiangiogenic medications were not discontinued after surgical treatment. As suggested by Staderini E. et al., a thorough understanding of information such as the patient’s underlying disease is a way to implement ethical treatment. This should be done prior to selecting a treatment method [13].

### 2.2. Data Collection

The study was conducted by analyzing clinical and radiological (panoramic radiograph) results. Items included in the analysis were as follows: (1) stage of disease, (2) type of antiangiogenic or antiresorptive agent, (3) administration method and duration, (4) cause of jaw necrosis, (5) location of jaw necrosis, (6) surgical treatment method, and (7) treatment results. Surgical treatment methods were divided into three categories: (1) Sequestrectomy during which only formed sequestrum was removed, (2) Saucerization involving removal of adjacent cortical bones and achieving blood circulation in a lesion when sequestrum formation is in doubt, but the boundaries are unclear, and (3) Curettage was performed when the boundaries of sequestrum were clear and surrounded by granulation tissue. When extensive lesions were present, they were removed with wide margins. Treatment rates were as follows; sequestrectomy 11.21% of cases (*n* = 13), saucerization 24.14% (*n* = 28), and curettage 64.66% (*n* = 75). Treatment results were categorized as: (1) successful when new epithelium formed in treated areas and no infection or clinical symptoms were observed. or as (2) failures when clinical symptoms recurred or infection occurred in treated area as indicated by: (1) lack of soft tissue continuity, (2) the presence of inflammatory exudate, (3) abscess, (4) pain, or (5) exposure of necrotic bone.

### 2.3. Statistical Analysis

Results are presented as frequencies and percentages for categorical variables and as means ± standard deviations (SD) for continuous variables. The significances of differences between participants’ characteristics were compared using Fisher’s exact test for categorical variables or the independent test or Mann–Whitney’s U test for continuous variables, as appropriate. The Shapiro–Wilk test was used to determine distribution normality. Logistic regression was performed to identify prognostic factors independently related to treatment outcomes. The statistical analysis was carried out using SPSS v.24.0 (IBM, New York, NY, USA), and *p* values of < 0.05 were considered statistically significant.

## 3. Results

### 3.1. Patients Factors

#### 3.1.1. Age and Gender

One hundred and sixteen patients (3 males and 113 females) met the study inclusion criteria. (total *n* = 116), indicating a considerable female predilection (97.4%; M/F ratio 1:38). Average patient age was 74.58 years (min 47, max 83, median 75).

#### 3.1.2. Antiresorptive Agents Type

Of the 116 study subjects, 102 took an antiresorptive agent (bisphosphonate); the drugs taken by the other 14 patients were unknown. The distribution of drug types taken by these 102 patients was as follows; pamidronate 6, alendronate 38, ibandronate 36, risedronate 18, and zoledronate 4.

#### 3.1.3. Antiresorptive Agent Dosing Methods and Durations

Average durations of oral and IV administrations were 5.81 ± 5.42 months (range 2 to 30 months) and 5.89 ± 5.64 months (range 1 to 10 months), respectively. The administration method was oral administration in 79.3% of cases (*n* = 92) and IV administration in 20.7% (*n* = 24).

#### 3.1.4. Causes of Necrosis

We investigated surgical causes of necrosis of the jaw and attributed these to extraction in 7.8% (9 cases), implant placement in 51.7% (60 cases), periodontal disease in 30.2% (35 cases), and denture stimulation in 10.3% (12 cases).

#### 3.1.5. Necrosis Locations

Lesions occurred in maxilla in 25% of cases (*n* = 29) and mandible in 75% of cases (*n* = 87); a maxillary to mandibular ratio of 1:3. Regarding jaw locations, lesions involved anterior teeth in 17.2% of cases (*n* = 20) and posterior teeth in 82.6% of cases (*n* = 96); an anterior to posterior ratio of 1:4.8.

#### 3.1.6. Treatment Results

Treatment success was achieved in 94.0% of the study subjects (109 cases; the success group), and treatment failure in 6.0% (7 cases; the failure group). Table 2 summarizes how patient factors influenced outcomes. However, the number of patients in the failure group was small (*n* = 7), and thus, overall results were not statistically significant. Nevertheless, zoledronate and IV injection were found to be associated with poor treatment results (*p* values < 0.1).

### 3.2. Stage Worsening

Six of the seven treatment failures were diagnosed as stage 2 and one as stage 3 at initial visits. At follow-up, none of the seven displayed continuous recovery of mucosa or inflammatory lesion reduction, and in all symptoms worsened. As a result, the six diagnosed as stage 2 progressed to stage 3, and in the stage 3 patient, lesions enlarged.

Univariate logistic regression analysis showed age (OR 0.90, 95% CI 0.82–0.98, *p* = 0.021), zoledronate administration (OR 21.40, 95% CI 2.48–184.71, *p* = 0.005), and IV administration (OR 4.99, 95% CI 1.04–23.87, *p* = 0.044) were associated with treatment failure (Table 3). However, multivariate logistic regression analysis using the backward elimination method showed that only the use of zoledronate (OR 21.40, 95% CI 2.48–184.71, *p* = 0.005) influenced results (Table 4).

Zoledronate administration (OR 21.40, 95% CI 2.48–184.71, *p* = 0.005) was found to be with poorer treatment results.

### 3.3. Times to Treatment Completion after Surgery

Patients were followed up at 10 days and three and six months after surgery. Times to bone healing were compared and evaluated based on panoramic radiograph examination results. When symptoms worsened without healing during follow-up, treatment was considered as a failure. Treatments for 37 patients were ended at three months postoperatively and for 53 patients at six months postoperatively. Of the 116 study subjects, 96 (82.8%) ended treatment within six months of surgery and 20 (17.2%) ended treatment at more than 6 months.

## 4. Discussion

This retrospective study was conducted to determine which treatment method is more suitable for MRONJ, the treatment of which continues to be controversial. AAMOS recommended MORNJ be treated as conservatively as possible in the Position Paper released in 2014 and in the update released in 2015 [2,6], because surgical treatment causes bone exposure, and thus, interferes with the goal of disease prevention. The Japanese Bone Metabolism Society (JSBMR) makes the same recommendation [7]. However, recent studies show that the prognosis of surgical treatment is better than that of conservative treatment [8]. For example, Hayashida et al. reported that healing was achieved in a high percentage of patients that underwent surgical treatment, that is, cure was achieved in 76.7% of 361 stage 2 MRONJ patients treated surgically but in only 25.2% of patients treated conservatively [9], and others have shown that wider surgery improves prognosis [10]. Eguchi et al. reported surgical treatment was successful in 89.3% patients but in only 33.3% of patients treated conservatively [14].

Our analysis of risk factors produced results that concurred with AAOMS position paper conclusion and showed a high rate of morbidity and high prevalence’s of a mandibular location and a female gender [2]. Our results concur, as surgical treatment of MRONJ in all stages achieved healing in 109 (94.0%) of the 116 study subjects and failed in only seven (6.0%). Six of the seven in the failure group with stage 2 MRONJ proceeded to stage 3, and in the other patient initially diagnosed as stage 3, lesions progressed, and subsequently, were surgically removed with adjacent areas. Fliefel et al. concluded that wider surgery was associated with better prognoses but cautioned that excessive surgery can be dangerous [7,10,15].

In the present study, follow-up checks were performed at 3 and 6 months postoperatively, and treatments were terminated when healing was confirmed. A dental panoramic examination was performed at each follow-up visit to confirm jaw recovery (whether bone was formed, etc.). As a result, treatment was discontinued in 82.6% of the study subjects during the first six postoperative months. This high rate of early termination reflects restoration of the oral environment, and thus, improved diet-related QOL.

The factors best known to be associated with MRONJ treatment outcomes are duration of antiangiogenic or antiresorptive agent use and drug type. Details of the ways medications affect the incidence of MRONJ are provided in the AAOMS 2014 position paper. In general, the incidence of MRONJ increases as the duration of drug administration increases [2], and is increased by antiresorptive agents like denosumab, which confer greater risk of MRONJ than BPs. However, we were not able to confirm this because our patients only received BPs. Nonetheless, because the incidence rate of MRONJ is BP type-dependent [16], we investigated the relation between BP type and treatment outcomes, and interestingly, we found that zoledronate (ZA) use may be associated with treatment failure. Many studies have reported that the use of ZA delays healing of tooth extraction sockets [17,18,19,20]. However, controversy exists regarding the nature of the relationship between total ZA dose and the incidence of MRONJ. For example, it has been reported that dosage, method of administration, and period of administration of ZA are not related to the occurrence of MRONJ [21]. On the other hand, there is a research result showing that use for 2.2 years is associated with MRONJ [22], and that the incidence of MRONJ increases with duration of ZA use [23]. It has been suggested that the reason why the properties of ZA differ from those of other BPs is due to the properties conferred by its nitrogen group [24], and it has been shown that its chemical properties enhance its abilities to bind to bone and inhibit bone resorption [24,25].

Drug holidays are also important when considering surgical intervention in patients with a history of taking antiangiogenic or antiresorptive agents. AAOMS argued that a drug holiday of about two months is appropriate based on consideration of BP half-lives [2]. However, Hasegawa et al. reported that removing the cause of infection in alveolar bone promptly without a drug holiday resulted in faster healing [26]. We did not analyze the effects of drug holidays and recommend that additional studies be conducted to determine the nature of the relation between them and treatment outcomes [17].

## 5. Limitation

The present study has a number of limitations that warrant consideration. First, it lacks information about administered BP doses, which are known to have a significant impact on the development of MRONJ [2]. Information such as smoking status, oral hygiene, socioeconomic status, health status, and presence of co-morbidities was not collected. Second, significance could not be shown due to the small size of the failure group. In addition, statistical errors cannot be considered because the total number of patients who visited the hospital from January 2014 to December 2018 was not calculated. Patini R. et al., reported that case reports negatively affect the journal’s impact factor [27]. Third, we did not consider drug holidays. Follow-up studies are needed to compensate for these deficiencies.

## 6. Conclusions

We investigated the success rates of surgical treatment in all stages of MRONJ and evaluated treatment outcomes using: (1) assessments of lesion exacerbation after treatment (defined as an AAOMS stage change), and (2) times to treatment termination. Of the 116 study subjects, 96 (83.0%) terminated treatment within 6 months of surgery, and 20 (17.0%) terminated treatment after 6 months or more. Treatment failure was observed in seven patients (6.0%). Six patients diagnosed with stage 2 advanced to stage 3, and stage 3 patients with enlarged lesions. So, active surgical intervention could greatly improve patient quality of life by shortening hospitalization periods and enabling early reconstruction of the oral environment.

## Figures and Tables

**Table 1 ijerph-17-08801-t001:** Various Types of Bisphosphonates (BPs) according to the official Name of BP.

Ingredient	Medicine					
**Alendronate**	Fosamax	Maxambil	Alenmax	Alendros	Alonate	Tevanate
	Fosaqueen	Daewoon Alendronate	Bonaid	Alenbon	Alenmax	Alont
	Posarend	Marend	Bisbon	Alrond	Allentop	Aidbone
	Posaronin	Alend	Alendro	Alenfos	Ginodron	Fosalon
**Zolendronate**	Aclasta inj.	Zometa inj.				
**Pamidronate**	Pamidronate					
**Risedronate**	Remabone	Ridroqun	Richbone	Ostol	Hutecs	Resenel
	Ridbon	Bontonel	Ostron	Risedronate	Actonel	Riseto
	Actoril	Kantonel	Daewon Resedronate	Risnel	Osnel	Hudron
**Ibandronate**	Bonviva					

**Table 2 ijerph-17-08801-t002:** Baseline and Clinical Characteristics of Patients in the two Groups.

Variable	Treatment Outcome		
	Success (No. 109)	Failure (No. 7)	*p* Value
**Sex**			
Female	107 (98.2%)	6 (85.7%)	0.172 ^1^
Male	2 (1.8%)	1 (14.3%)	
**Age, years**	75.03 ± 7.00	67.57 ± 14.21	0.216 ^2^
**Stage**			
1	7 (6.4%)	0 (0.0%)	1.000 ^1^
2	81 (74.3%)	6 (85.7%)	
3	21 (19.3%)	1 (14.3%)	
**Occurrence of jaw**			
Mandible	82 (75.2%)	5 (71.4%)	1.000 ^1^
Maxilla	27 (24.8%)	2 (28.6%)	
**Occurrence of location**			
Posterior teeth	88 (81.5%)	6 (100.0%)	0.588 ^1^
Anterior teeth	20 (18.5%)	0 (0.0%)	
**Drug types**			
Alendronate	37 (33.94%)	1 (14.29%)	0.424 ^1^
Risedronate	16 (14.68%)	2 (28.57%)	0.322 ^1^
Zoledronate	2 (1.83%)	2 (28.57%)	0.018 ^1^
Ibandronate	35 (32.11%)	1 (14.29%)	1.000 ^1^
Pamidronate	5 (4.59%)	1 (14.29%)	0.361 ^1^
unkown	14 (12.84%)		
**Administrative route**			
IV (injection)	23 (21.1%)	4 (57.1%)	0.0501 ^1^
PO (per oral)	89 (81.7%)	4 (57.1%)	0.139 ^1^
**Period of medication (years)**	5.98 ± 5.54 (med 4.5)	3.29 ± 2.06 (med 3)	0.230 ^3^
**Causes of necrosis**			
Implantation	9 (8.3%)	0 (0.0%)	0.925 ^1^
Extraction	55 (50.5%)	5 (71.4%)	
Perioperative lesion	33 (30.3%)	2 (28.6%)	
Denture irritation	12 (11.0%)	0 (0.0%)	
**Treatment**			
Curettage	70 (64.2%)	5 (71.4%)	1.000 ^1^
Saucerization	26 (23.9%)	2 (28.6%)	
Sequestrectomy	13 (11.9%)	0 (0.0%)	

Values are either frequencies with percentages in parentheses or means ± standard deviations. ^1^
*p* values were obtained using Fisher’s exact test. ^2^
*p* values were obtained using the independent *t*-test. ^3^
*p* values were obtained using the Mann–Whitney U test. Shapiro–Wilk’s test was used to determine normality.

**Table 3 ijerph-17-08801-t003:** Risk Factor analysis by univariate logistic Regression.

Variable	OR	95% CI	*p* Value
**Sex**			
Female	Ref.		0.091
Male	8.92	0.71–112.77	
**Age, years**	0.90	0.82–0.98	0.021
**Stage**			
1	Ref.		N/E
2	N/E		
3	N/E		
**Occurrence of jaw**			
Mandible	Ref.		0.822
Maxilla	1.21	0.22–6.63	
**Occurrence of location**			
Posterior teeth	Ref.		N/E
Anterior teeth	N/E		
**Drug types**			
Alendronate	0.32	0.04–2.79	0.306
Risedronate	2.16	0.39–12.08	0.379
Zoledronate	21.40	2.48–184.71	0.005
Ibandronate	0.81	0.15–4.39	0.808
Pamidronate	2.86	0.30–27.73	0.364
**Administrative route**			
IV (injection)	4.99	1.04–23.87	0.044
PO (per oral)	0.30	0.06–1.45	0.133
**Period of medication (years)**	0.82	0.61–1.11	0.200
**Causes of necrosis**			
Implantation	Ref.		N/E
Extraction	N/E		
Perioperative lesion	N/E		
Denture irritation	N/E		
**Treatment**			
Curettage	Ref.		
Saucerization	1.08	0.20–5.90	0.932
Sequestrectomy	N/E		N/E

Age (OR 0.90, 95% CI 0.82–0.98, *p* = 0.021), taking zoledronate (OR 21.40, 95% CI 2.48–184.71, *p* = 0.005), and IV administration (OR 4.99, 95% CI 1.04–23.87, *p* = 0.044) were found to have significant effects on treatment results. N/E: Not estimable because of no patient in the failure group.

**Table 4 ijerph-17-08801-t004:** Result of multivariate logistic regression analysis using the backward elimination method.

Variable	OR	95%CI	*p* Value
zoledronate	21.40	2.48–184.71	0.005

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
