# Peer review of "Surgical Treatment of Medication-Related Osteonecrosis of the Jaw: A Retrospective Study"

_ijerph, 2020, doi:10.3390/ijerph17238801_

Round 1

Reviewer 1 Report

TITLE: please indicate the type of study

INTRODUCTION: can you rephrase the aim of the study, please? What does it mean time to treatment termination?

MATERIALS AND METHODS: 

page 2, line 49: were the patients consecutively enrolled?

did you calculate the sample size?

can you explain some characteristics of the participants (smoking status, health status)

Did you prescribe any antibiotic therapy?

RESULTS:

page 2, lines 85-86: it is possible to know the drug dosage?

DISCUSSION:

please add a paragraph with "limitations and strenghts" of the study

page 6, line 189: please add a reference for this sentence; as a suggestion, please find a study below:

Staderini E, De Luca M, Candida E, Rizzo MI, Rajabtork Zadeh O, Bucci D, Zama M, Lajolo C, Cordaro M, Gallenzi P. Lay People Esthetic Evaluation of Primary Surgical Repair on Three-Dimensional Images of Cleft Lip and Palate Patients. Medicina (Kaunas). 2019 Sep 8;55(9):576. doi: 10.3390/medicina55090576. PMID: 31500380; PMCID: PMC6780772.

Author Response

TITLE: please indicate the type of study

- I changed the title."Surgical treatment of medication-related osteonecrosis of the jaw patients"  

INTRODUCTION: can you rephrase the aim of the study, please? What does it mean time to treatment termination?

- An additional explanation of treatment termination was described. "Termination means that the patient does not show any abnormalities such as pain or inflammation and restores the normal continuity of the oral mucosa."

MATERIALS AND METHODS: 

page 2, line 49: were the patients consecutively enrolled?

- Yes. Patients visited the hospital continuously during the period. However, there is no constant periodicity.

did you calculate the sample size?

- Total number is 116.

can you explain some characteristics of the participants (smoking status, health status)

Unfortunately, nothing other than the gender and age information described in the paper was collected.

Did you prescribe any antibiotic therapy?

- Information on the antibiotics used is further described."These patients were prescribed antibiotics for two weeks prior to surgical treatment. As for antibiotics, 500 mg of amoxacillin and chlorhexidine gargle were routinely used, and when pus was observed, 250 mg of metronidazole was additionally used."

RESULTS:

page 2, lines 85-86: it is possible to know the drug dosage?

Active inquiries were made to the clinic who prescribed BPs, but they were reluctant to provide detailed information. I am very sorry for this too.

DISCUSSION:

please add a paragraph with "limitations and strenghts" of the study

- I added what you said at the end of the discussion.

page 6, line 189: please add a reference for this sentence; as a suggestion, please find a study below:

Staderini E, De Luca M, Candida E, Rizzo MI, Rajabtork Zadeh O, Bucci D, Zama M, Lajolo C, Cordaro M, Gallenzi P. Lay People Esthetic Evaluation of Primary Surgical Repair on Three-Dimensional Images of Cleft Lip and Palate Patients. Medicina (Kaunas). 2019 Sep 8;55(9):576. doi: 10.3390/medicina55090576. PMID: 31500380; PMCID: PMC6780772.

- The paper was cited."E. Staderini et al. tried to analyze the facial hair of patients with cleft lip in 3D. In a subsequent study of this study, using the 3D technology, an attempt to analyze the alveolar bone recovery pattern of MRONJ patients who received surgical treatment will also show good results"

Reviewer 2 Report

The current paper aimed to elucidate the effectiveness of surgical treatment for medication-related osteonecrosis of the jaw. The study included patients under bisphosphonate treatment who experienced osteonecrosis and subsequently were treated surgically. The results of this study could support future decision making during the evaluation of the treatment options of MRONJ patients.

Broad comments:

The title suggests that the study focuses on elderly people, whereas patients from the age of 47 to 83 were included. Hence the title does not fit the investigated population. There is also not a specific comment or section focusing on MRONJ in the elderly.

Section 2.1. Patients: were there any inclusion or exclusion criteria concerning age, race, preexisting diseases (e.g. bone cancer)?

Section 3.1.3. Antiangiogenic and antiresorptive agent dosing methods and durations: how does this section agree with the statement of the authors in the discussion section that all included patients only received bisphosphonates? Did the patients also receive antiangiogenic and/or antiresorptive drugs and if yes, which ones?

Section 3.1.6. Surgical treatment method: the first part of this section (lines 103-105) is a pure repetition of the text in section 2.2. (lines 59-63) and therefore redundant.

Section 3.1.7. Treatment results: the first part of this section (lines 109-111) is a repetition of the text in section 2.2. (lines 64-67) and therefore redundant.

In the discussion section, the authors list their own results and those of other studies, but without directly comparing them, which would be the main goal of a discussion.

It would also be interesting to compare the high rate of early treatment termination after surgery with known conservative treatment results.

In the current study, zoledronate was associated with a higher risk for treatment failure. Are there any other studies that reported the same or similar results?

About a quarter of the discussion is about drug holidays, which are an interesting topic to investigate, but since the effect of drug holidays was not analyzed in the current study, this topic should not be addressed in such a vast size.

A limitations section is missing in the current manuscript and should discuss the validity of zoledronate as a risk factor since the case number was too small to make conclusions with certainty. Furthermore, the overall low failure number, which is of course pleasant for patients and physicians, reduces the power of any conclusions regarding risk factors. Hence, for significant results, a higher number of patients would be needed.

Specific comments:

Lines 22-25: the reference for the definition of MRONJ is missing.

Lines 39: The mentioned “many studies” should be referenced.

Line 66-67: please define the “clinical symptoms” that led to treatment termination as a failure.

Line 81: was the age normally distributed or log-normal? Please indicate the median for the participant’s age.

Line 84: if the drug was unknown, how can the authors be sure of any causality with osteonecrosis? Was this considered in the calculations and statistics shown in Table 1?

Line 91: please also indicate the shortest administration period. Was there also mixed administration (IV and oral)?

Table 1: the percentages of the failure outcomes under “Drug types” is 114.4% when totalized. How is this possible? In contrast, the success outcomes totalize at 89.8%.

Table 1: under “Administrative route” please complement “Injection” with “IV”, which was used in the text.

Table 1: for “Period of medication (years)” please also indicate the median duration.

Table 1 footnotes: “N/E: Not estimable because of no patient in the failure group.” does not apply for table 1 but table 2, where it is missing in the footnotes.

Table 2: please explain how the reference among the subgroups (e.g. female for sex) was chosen and how it was calculated.

Table 2: under “Administrative route” please complement “Injection” with “IV”, which was used in the text.

Table 2: under “Cause of necrosis” in table 1 there were 5 failure outcomes for extraction, while in table 2 there are no patients for the same subgroup.

Line 139: the verb “be” is missing in the sentence “was found to significantly and independently associated with poorer treatment results”.

Line 156: please add a reference for the “others”.

Line 158: what is meant by “aggressive surgical treatment”? The term “aggressive” has not been used in this context so far.

Line 165: write out in full OPTG.

Author Response

The title suggests that the study focuses on elderly people, whereas patients from the age of 47 to 83 were included. Hence the title does not fit the investigated population. There is also not a specific comment or section focusing on MRONJ in the elderly.

- The title was modified.

Section 2.1. Patients: were there any inclusion or exclusion criteria concerning age, race, preexisting diseases (e.g. bone cancer)?

Patients who took steroids such as rheumatoid arthritis were excluded, but there were no other exclusion criteria.

The following content was added."Exclusion criteria are as follows. 1) Patients with a follow-up period of less than 3 months. 2) Patients with a history of using steroids for disease treatment." 

Section 3.1.3. Antiangiogenic and antiresorptive agent dosing methods and durations: how does this section agree with the statement of the authors in the discussion section that all included patients only received bisphosphonates? Did the patients also receive antiangiogenic and/or antiresorptive drugs and if yes, which ones?

It was a misleading title selection. To correct this error, "antiangiogenic" was excluded from the title.

Section 3.1.6. Surgical treatment method: the first part of this section (lines 103-105) is a pure repetition of the text in section 2.2. (lines 59-63) and therefore redundant.

To avoid repetition, unnecessary sentences were removed.

Section 3.1.7. Treatment results: the first part of this section (lines 109-111) is a repetition of the text in section 2.2. (lines 64-67) and therefore redundant.

Deleted the repeated part.

In the discussion section, the authors list their own results and those of other studies, but without directly comparing them, which would be the main goal of a discussion.

It would also be interesting to compare the high rate of early treatment termination after surgery with known conservative treatment results.

- I have added the content."Hayashida et al. found cure in 76.7% of 361 stage 2 MRONJ patients who received surgical treatment, but only 25.2% of patients treated conservatively. Also, Eguchi et al. Concluded that surgical treatment was more effective in 25 patients. They reported that treatment was successful in 89.3% of patients receiving surgical treatment and 33.3% of patients receiving non-surgical treatment."

In the current study, zoledronate was associated with a higher risk for treatment failure. Are there any other studies that reported the same or similar results?

No related papers have been found indicating that ZA is directly related to lesion recurrence. However, many studies have shown that surgical approaches, such as extraction, increase the incidence of MRONJ in patients taking ZA. I added about the content.

About a quarter of the discussion is about drug holidays, which are an interesting topic to investigate, but since the effect of drug holidays was not analyzed in the current study, this topic should not be addressed in such a vast size.

- Some of the content related to drug holiday has been deleted.

A limitations section is missing in the current manuscript and should discuss the validity of zoledronate as a risk factor since the case number was too small to make conclusions with certainty. Furthermore, the overall low failure number, which is of course pleasant for patients and physicians, reduces the power of any conclusions regarding risk factors. Hence, for significant results, a higher number of patients would be needed.

- Contents on the limitations of this study were added.

Specific comments:

Lines 22-25: the reference for the definition of MRONJ is missing.

- The description of the definition is as follows, and has been added.

" It refers to ischemic necrosis of the jaw caused by drugs that inhibit angiogenesis (antiangiogenic agents) and inhibit bone metabolism (antiresorptive agents) which are used to treat osteoporosis."

Lines 39: The mentioned “many studies” should be referenced.

mentioned as a reference.

Line 66-67: please define the “clinical symptoms” that led to treatment termination as a failure.

- The description has been added. "Treatment failure was considered when the following symptoms were observed: 1) destruction of soft tissue continuity, 2) occurrence of inflammatory exudate, 3) abscess, 4) pain, 5) exposure of necrotic bone."

Line 81: was the age normally distributed or log-normal? Please indicate the median for the participant’s age.

- median age is 75.

Line 84: if the drug was unknown, how can the authors be sure of any causality with osteonecrosis? Was this considered in the calculations and statistics shown in Table 1?

Patients whose drug type has not been identified are those who clearly remember the history of osteoporosis or the duration of the drug, but do not know the type of drug. The case of taking vitamin D was not included in the subject of this study. Therefore, patients with unknown drugs were also included in the statistics.

Line 91: please also indicate the shortest administration period. Was there also mixed administration (IV and oral)?

- (min 2, max 30). The mixed dose period was not considered.

Table 1: the percentages of the failure outcomes under “Drug types” is 114.4% when totalized. How is this possible? In contrast, the success outcomes totalize at 89.8%.

- There seems to be an error in writing. Modified.

Table 1: under “Administrative route” please complement “Injection” with “IV”, which was used in the text.

- Modified.

Table 1: for “Period of medication (years)” please also indicate the median duration.

- It was added.

Table 1 footnotes: “N/E: Not estimable because of no patient in the failure group.” does not apply for table 1 but table 2, where it is missing in the footnotes.

- It was corrected.

Table 2: please explain how the reference among the subgroups (e.g. female for sex) was chosen and how it was calculated.

The calculation for the given example is as follows. "ORmale vs. female = 1x107/2x6 = 8.92"

For each item, the standard (ref. in Table 2) was set and calculated as a ratio.

Table 2: under “Administrative route” please complement “Injection” with “IV”, which was used in the text.

- It was changed.

Table 2: under “Cause of necrosis” in table 1 there were 5 failure outcomes for extraction, while in table 2 there are no patients for the same subgroup.

The value for this item is N/E.(It is not empty). This occurs because the value of the ref. "implantation" is "0" and cannot be calculated.

Line 139: the verb “be” is missing in the sentence “was found to significantly and independently associated with poorer treatment results”.

- It was added.

Line 156: please add a reference for the “others”.

- Reference was added.

Line 158: what is meant by “aggressive surgical treatment”? The term “aggressive” has not been used in this context so far.

- I removed that word.

Line 165: write out in full OPTG.

- It was changed to "dental panoramic"

Reviewer 3 Report

The issue is interesting and the manuscript is well written, but I have some suggestions:

The authors should remove the reference (Ruggiero et al., 2014) in the abstract, write the meaning of OPTG (discussion section) and include updated references.

Finally, the authors should explain why zoledronate was associated with treatment failure. Including a paragraph about this the discussion section will be improved. 

Author Response

The issue is interesting and the manuscript is well written, but I have some suggestions:

The authors should remove the reference (Ruggiero et al., 2014) in the abstract, write the meaning of OPTG (discussion section) and include updated references.

- I have corrected the point.

Finally, the authors should explain why zoledronate was associated with treatment failure. Including a paragraph about this the discussion section will be improved.

- I have added the content. "Many studies have reported that the use of ZA delays healing of the tooth extraction socket. However, there is controversy over the relationship between the total dose of ZA and the incidence of MRONJ. There is a result that the dosage, method of administration, and period of administration of ZA are not related to the occurrence of MRONJ. On the other hand, there is a research result showing that use for 2.2 years causes MRONJ. In addition, there is a research result that the incidence of MRONJ increases as the period of use increases. The reason that ZA has different properties from other BPs is due to the properties of the nitrogen group of ZA. The chemical properties of ZA allow this drug to bind better to bone and have a strong bone resorption inhibition."

Round 2

Reviewer 1 Report

TITLE: please indicate the type of study

ABSTRACT: 

page 1, lines 9-12: this paragraph should be removed, as it is too generic and not strictly necessary for the content of the paper

INTRODUCTION: page 1, lines 28-33: this paragraph is too generic. You should briefly explain the treatment workflow proposed by AAOMFS and define the meaning of "conservative treatment"

page 1, lines 44-45: you should remove this sentence.

MATERIALS AND METHODS:

Sample size calculation, ethical board approval are missing.

Were the patients consecutively enrolled?

Some information is missing (smoking status, oral hygiene, socioeconomic status, health status, presence of co-morbidities)

page 2, line 52: please add a table with the types medications (per-oral or intravenous) for osteoporosis and specify that the medications dose are unknown, as mentioned in the discussion.

page 2, lines 52-53: please specify the AAOMS diagnostic criteria for medication-related ostheonecrosis of the jaw

page 2, line 55: please add a reference to this sentence; as a suggestion, please find a study below:

Manage Impacted Third Molars: Germectomy or Delayed Removal? A Systematic Literature Review. Medicina (Kaunas). 2019 Mar 26;55(3):79. doi: 10.3390/medicina55030079. PMID: 30917605; PMCID: PMC6473914.

Page 2, line 60: you need to move here the paragraph from page 3, lines 104-107, as you need to point out how many subjects performed sequestrectomy, saucerization and curettage? Which diagnostic criteria influence your surgical protocol (it is not clear the choice between sequestrectomy and curettage)?

RESULTS:

the results secton should be organized to answer the research question.

page 5, line 141: "as a failure" instead of "to have failed"

page 5, lines 142-143: please rephrase this sentence. What does it mean "treatment were complete"?

DISCUSSION:

page 7, line 198: please add a reference to this sentence; as a suggestion, please find a study below:

Patini R, Staderini E, Camodeca A, Guglielmi F, Gallenzi P. Case Reports in Pediatric Dentistry Journals: A Systematic Review about Their Effect on Impact Factor and Future Investigations. Dent J (Basel). 2019 Oct 24;7(4):103. doi: 10.3390/dj7040103. PMID: 31652916; PMCID: PMC6960525.

CONCLUSIONS:

the conclusion should answer the research question: "we investigated the success rates of surgical treatment in all stages of MRONJ and evaluated treatment outcomes using: 1) assessments of lesion exacerbation after treatment (defined as an AAOMS stage change), and 2) times to treatment termination"

REFERENCES:

you should remove the old references from the manuscript.

Author Response

TITLE: please indicate the type of study

- Added "retrospective study"

ABSTRACT: 

page 1, lines 9-12: this paragraph should be removed, as it is too generic and not strictly necessary for the content of the paper

- It was removed.

INTRODUCTION: page 1, lines 28-33: this paragraph is too generic. You should briefly explain the treatment workflow proposed by AAOMFS and define the meaning of "conservative treatment"

- "When treating patients in stage 1 and 2, they recommended conservative treatment - taking antibiotics and trying gargle with 0.12% chlorhexidine first. When there is a concern about colonization of bacteria due to the formation of sequestra, a surgical procedure to selectively remove them was recommended. "

page 1, lines 44-45: you should remove this sentence.

- In fact, this is another reviewer's request. However, it was deleted from the modified manuscript.

MATERIALS AND METHODS:

Sample size calculation, ethical board approval are missing.

- IRB number and sample size calculation are specified."Institutional Review Board of Pusan National University Dental Hospital (PNUDH-2019-034)", "The total number of patients who satisfied the conditions was 116."

Were the patients consecutively enrolled?

- "They enrolled continuously during the period, but there was no regular periodicity."

Some information is missing (smoking status, oral hygiene, socioeconomic status, health status, presence of co-morbidities)

- "Information such as smoking status, oral hygiene, socioeconomic status, health status, and presence of co-morbidities was not collected."

page 2, line 52: please add a table with the types medications (per-oral or intravenous) for osteoporosis and specify that the medications dose are unknown, as mentioned in the discussion.

- The types of drugs are summarized and specified in the table 1. In addition, the fact that the identification of the drug dosage was unclear was also specified.

page 2, lines 52-53: please specify the AAOMS diagnostic criteria for medication-related ostheonecrosis of the jaw

- "; 1) Current or previous treatment with antiresorptive or antiangiogenic agents; 2) Exposed bone or bone that can be probed through an intraoral or extraoral fistula(e) in the maxillofacial region that has persisted for more than eight weeks; and 3) No history of radiation therapy to the jaws or obvious metastatic disease to the jaws."

page 2, line 55: please add a reference to this sentence; as a suggestion, please find a study below:

Manage Impacted Third Molars: Germectomy or Delayed Removal? A Systematic Literature Review. Medicina (Kaunas). 2019 Mar 26;55(3):79. doi: 10.3390/medicina55030079. PMID: 30917605; PMCID: PMC6473914.

- The following sentence was added. "As suggested by E. Staderini et al., a thorough understanding of information such as the patient's underlying disease is a way to implement ethical treatment. This should be done prior to selecting a treatment method." 

Page 2, line 60: you need to move here the paragraph from page 3, lines 104-107, as you need to point out how many subjects performed sequestrectomy, saucerization and curettage? Which diagnostic criteria influence your surgical protocol (it is not clear the choice between sequestrectomy and curettage)?

- I moved the paragraph to page 2, and I added the following: "2) Saucerization involving removal of adjacent cortical bones and achieving blood circulation in a lesion when sequestrum formation is in doubt, but the boundaries are unclear, and 3) Curettage was performed when the boundaries of sequestrum were clear and surrounded by granulation tissue. "

RESULTS:

the results secton should be organized to answer the research question.

- The section of the Result section was intended to investigate the effectiveness of surgical treatment, including treatment results and stage worsening. It also showed that it took a short time to complete treatment, including Times to treatment completion after surgery.

page 5, line 141: "as a failure" instead of "to have failed"

- Corrected.

page 5, lines 142-143: please rephrase this sentence. What does it mean "treatment were complete"?

- changed to "ended" instead of "complete".

DISCUSSION:

page 7, line 198: please add a reference to this sentence; as a suggestion, please find a study below:

Patini R, Staderini E, Camodeca A, Guglielmi F, Gallenzi P. Case Reports in Pediatric Dentistry Journals: A Systematic Review about Their Effect on Impact Factor and Future Investigations. Dent J (Basel). 2019 Oct 24;7(4):103. doi: 10.3390/dj7040103. PMID: 31652916; PMCID: PMC6960525.

- The following sentence was added. "Patini R. et al. reported that case reports negatively affect the journal's impact factor."

CONCLUSIONS:

the conclusion should answer the research question: "we investigated the success rates of surgical treatment in all stages of MRONJ and evaluated treatment outcomes using: 1) assessments of lesion exacerbation after treatment (defined as an AAOMS stage change), and 2) times to treatment termination"

- "we investigated the success rates of surgical treatment in all stages of MRONJ and evaluated treatment outcomes using: 1) assessments of lesion exacerbation after treatment (defined as an AAOMS stage change), and 2) times to treatment termination. Of the 116 study subjects, 96 (82.76%) terminated treatment within 6 months of surgery, and 20 (17.24%) terminated treatment after 6 months or more. Treatment failure was observed in 7 patients. Six patients diagnosed with stage 2 advanced to stage 3, and stage 3 patients with enlarged lesions. "

REFERENCES:

you should remove the old references from the manuscript.

- I removed old references.

Reviewer 2 Report

Most issues have been adjusted appropriately. However, some points concerning the discussion section have remained unsolved:

Reviewer: Line 158: what is meant by “aggressive surgical treatment”? The term “aggressive” has not been used in this context so far.

Author answer: - I removed that word.

Reviewer: the word has not been removed, even though the authors had stated so.

Reviewer: the authors have added a limitations section as requested where they correctly stated “Second, significance could not be shown due to the small size of the failure group.” However, during the discussion, the authors still refer to the significant results regarding zoledronate, even though it is stated otherwise in the limitations section. Hence, during the discussion, the term “significant association” should not be used regarding the results of zoledronate as a risk factor.

Author Response

The word you pointed out was deleted. In addition, the discussion on the association between ZA and treatment failure was revised.